# Enhanced superconductivity and coexisting ferroelectricity at oxide interfaces

Meng Zhang [1,6] ✉, Ming Qin [1,6], Yanqiu Sun[1,6], Siyuan Hong[1], Yi Zhou [2] & Yanwu Xie [1,3,4,5] ✉

The coexistence of superconductivity and ferroelectricity is rare due to their conflicting requirements: superconductivity relies on free charge carriers, whereas ferroelectricity typically occurs in insulating systems. At $LaAlO_3$/ $KTaO_3$ interfaces, we demonstrate the coexistence of two-dimensional superconductivity and ferroelectricity, enabled by the unique properties of $KTaO_3$ as a quantum paraelectric. Systematic gating and poling experiments reveal an enhancement of the superconducting transition temperature ($T_c$) by ~0.2–0.6 K and bistable transport properties, including hysteresis, strongly suggesting the existence of switchable ferroelectric polarization in the interfacial conducting layer. Raman scattering measurements and hysteresis loops indicate robust ferroelectricity below 50 K. The $T_c$ enhancement is attributed to ferroelectric polarization-induced reduction in dielectric constant, which narrows the interfacial potential well, confining carriers closer to the interface. The bistability arises from switchable ferroelectric polarization, which modulates the potential well depending on polarization direction. These findings establish a straightforward mechanism coupling ferroelectricity and superconductivity, providing a promising platform for exploring their interplay.

The coexistence of superconductivity and ferroelectricity represents a long-standing challenge due to their fundamentally conflicting requirements. Superconductivity demands on a high density of free charge carriers, while ferroelectricity is generally found in insulating materials. The introduction of free charges into such materials screens long-range Coulomb interactions, suppressing ferroelectricity. Although the coexistence of free carriers and polarization has been proposed[1] and observed[2,3] in polar metals, including polar superconductors[4,5], the polarization in these systems is non-switchable. Recently, however, superconductivity and ferroelectricity were observed to coexist in two-dimensional (2D) van der Waals heterostructures[6], where ferroelectricity arises from mechanisms distinct from conventional Coulomb interactions.

Oxide interfaces, particularly those based on $SrTiO_3$ (STO)[7–9] and $KTaO_3$ (KTO)[10–15], offer a compelling alternative for exploring the interplay between superconductivity and polarization. Both STO and KTO are wide-gap semiconductors (3.2 eV for STO and 3.6 eV for KTO) and quantum paraelectric[16,17]. Electron-doped STO was the first oxide superconductor discovered[18], and its proximity to ferroelectricity inspired the discovery of high-temperature cuprate superconductors[19]. Quantum ferroelectric fluctuations have been proposed as a potential pairing mechanism in these systems[20–27], and a ferroelectric quantum phase transition was observed inside the superconducting dome of STO[28]. Additionally, when STO is combined with insulating oxides such as $LaAlO_3$ (LAO), 2D superconductivity emerges at the interface[8]. In STO-based interfaces, superconducting-ferroelectric coexistence has been achieved via $^{18}O$ substitution[19], and ferroelectric-switchable 2D electron gases have been realized through Ca alloying[29]. Furthermore, the effects of ferroelectric/ferroelastic domains on both conductivity[30,31] and superconducting pairing[32] have

[1]School of Physics, and State Key Laboratory for Extreme Photonics and Instrumentation, Zhejiang University, Hangzhou, China. [2]Institute of Physics, Chinese Academy of Sciences, Beijing, China. [3]College of Optical Science and Engineering, Zhejiang University, Hangzhou, China. [4]Collaborative Innovation Center of Advanced Microstructures, Nanjing University, Nanjing, China. [5]Hefei National Laboratory, Hefei, China. [6]These authors contributed equally: Meng Zhang, Ming Qin, Yanqiu Sun. ✉e-mail: physmzhang@zju.edu.cn; ywxie@zju.edu.cn

been suggested. However, the full integration of superconductivity with electrically switchable ferroelectric polarization, despite being conceptually proposed[29], has remained experimentally unrealized.

In contrast, while superconductivity has not been observed in electron-doped bulk KTO[33], recent studies reveal that 2D superconductivity can emerge at KTO surfaces[34,35] and interfaces[11–13]. Notably, KTO-based systems exhibit significantly higher superconducting transition temperature ($T_c$) than STO-based interfaces, making them more promising for exploring the interplay between ferroelectric polarization and superconductivity. Here, we demonstrate the coexistence of superconductivity and ferroelectricity at LAO/KTO interfaces, providing a novel platform for studying this unconventional phenomenon.

## Results

### Universally enhanced superconductivity and bistability
A noteworthy feature of KTO interface superconductivity is its tunability via gate bias ($V_G$) across KTO[13,36–38], as illustrated in Fig. 1a. Over

the past several years, we have gated more than 100 LAO/KTO samples and identified a universal phenomenon: after gating experiments (where a $V_G$ of up to ± 200 V was applied and subsequently removed, referred to as "poling"), the $T_c$ of LAO/KTO exhibited a significant enhancement of ~0.2–0.6 K (see Fig. 1b for selected results). Additionally, the normal-state sheet resistance ($R_s$) exhibits a bistable characteristic after poling with different bias polarities (Fig. 1c), indicating the presence of ferroelectricity.

### Ferroelectric hysteresis under gating cycles
To further investigate these phenomena, we examined the transport behaviors of a typical LAO/KTO(111) Hall bar device under continuous gating cycles. For each $V_G$ value, both the temperature dependence of $R_s(T)$ and the Hall effect (measured at $T = 4.5$ K) were recorded. Figure 2a shows the $R_s(T)$ curve before any $V_G$ was applied (denoted as "origin"). A metallic behavior, followed by a superconducting transition at $T_c = 1.89$ K (defined as the temperature where $R_s$ drops to 50% of

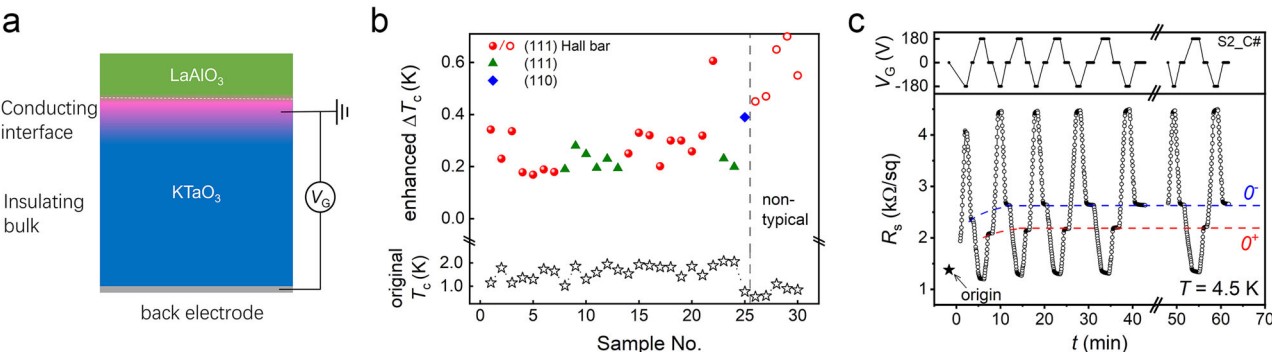

**Fig. 1 | Universally enhanced superconductivity and bistability. a** Schematic illustration of the gating setup for the LAO/KTO interface, with $V_G$ applied across the KTO substrate. The polarity of $V_G$ is defined relative to the interface. "Poling" refers to the process of applying $V_G$ and then removing it. **b** Enhancement of $T_c$ ($\Delta T_c$) for multiple LAO/KTO samples after poling with $V_G = -180$ V. Open black stars: original $T_c$. Closed blue diamonds: unpatterned LAO/KTO(110) samples. Closed green triangles: unpatterned LAO/KTO(111) samples. Closed red circles: LAO/

KTO(111) Hall bar devices. Open red circles: LAO/KTO(111) Hall bar devices where LAO films were deposited using a non-typical procedure (details provided in **Methods**). **c** Time-dependent sheet resistance ($R_s$) at $T = 4.5$ K measured while $V_G$ was switched repeatedly between 0, −180, 0, +180 V. "Origin" denotes the state before any $V_G$ was applied. At $V_G = 0$, two distinct $R_s$ states are observed: "$O^+$" and "$O^-$", corresponding to the states after removing positive or negative $V_G$, respectively.

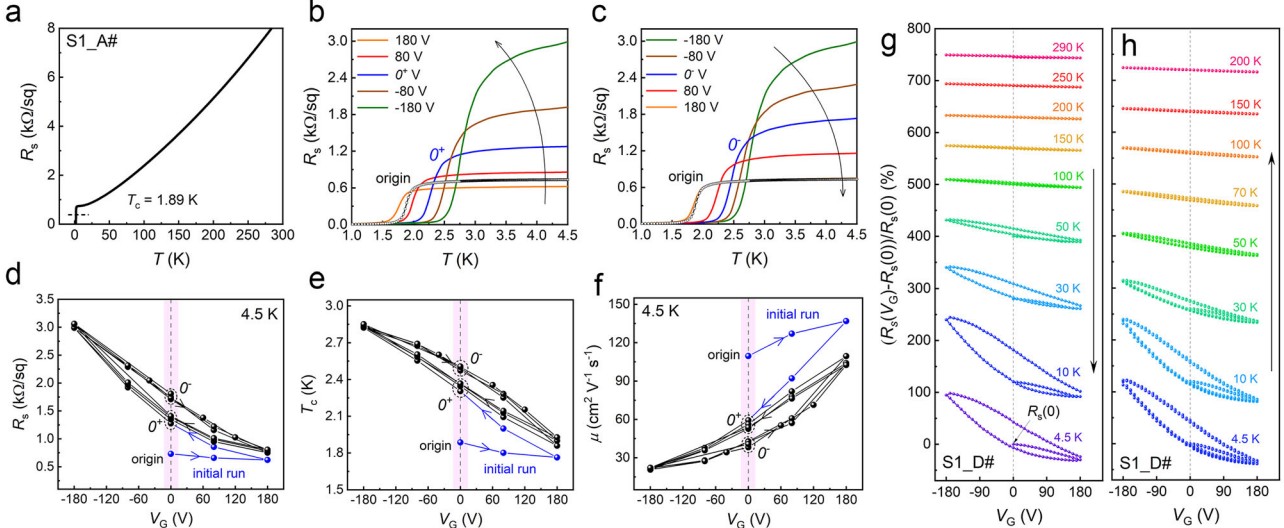

**Fig. 2 | Ferroelectric hysteresis at the LAO/KTO interface.** Consecutive gating cycles between $V_G = +180$ V and −180 V were performed on a typical LAO/KTO(111) Hall bar device (S1_A#): **a** Temperature-dependent $R_s(T)$ curve for the "origin" state (before any gating). **b, c,** Temperature-dependent $R_s(T)$ curves during a single gating cycle: **b** sweeping $V_G$ from +180 V to −180 V; **c** sweeping $V_G$ from −180 V to

+180 V. **d, e, f** Hysteresis loops during consecutive gating cycles: **d** $R_s$-$V_G$; **e,** $T_c$-$V_G$; **f,** $\mu$-$V_G$. The pink shading highlights the $V_G = 0$ region. **g, h** Evolution of $R_s$-$V_G$ loops with temperature: **g** measured in decreasing temperature order; **h** measured in increasing temperature order. To improve clarity, the normalized form [$R_s(V_G)$-$R_s(0)$]/$R_s(0)$ is used, with curves shifted vertically for better visualization.

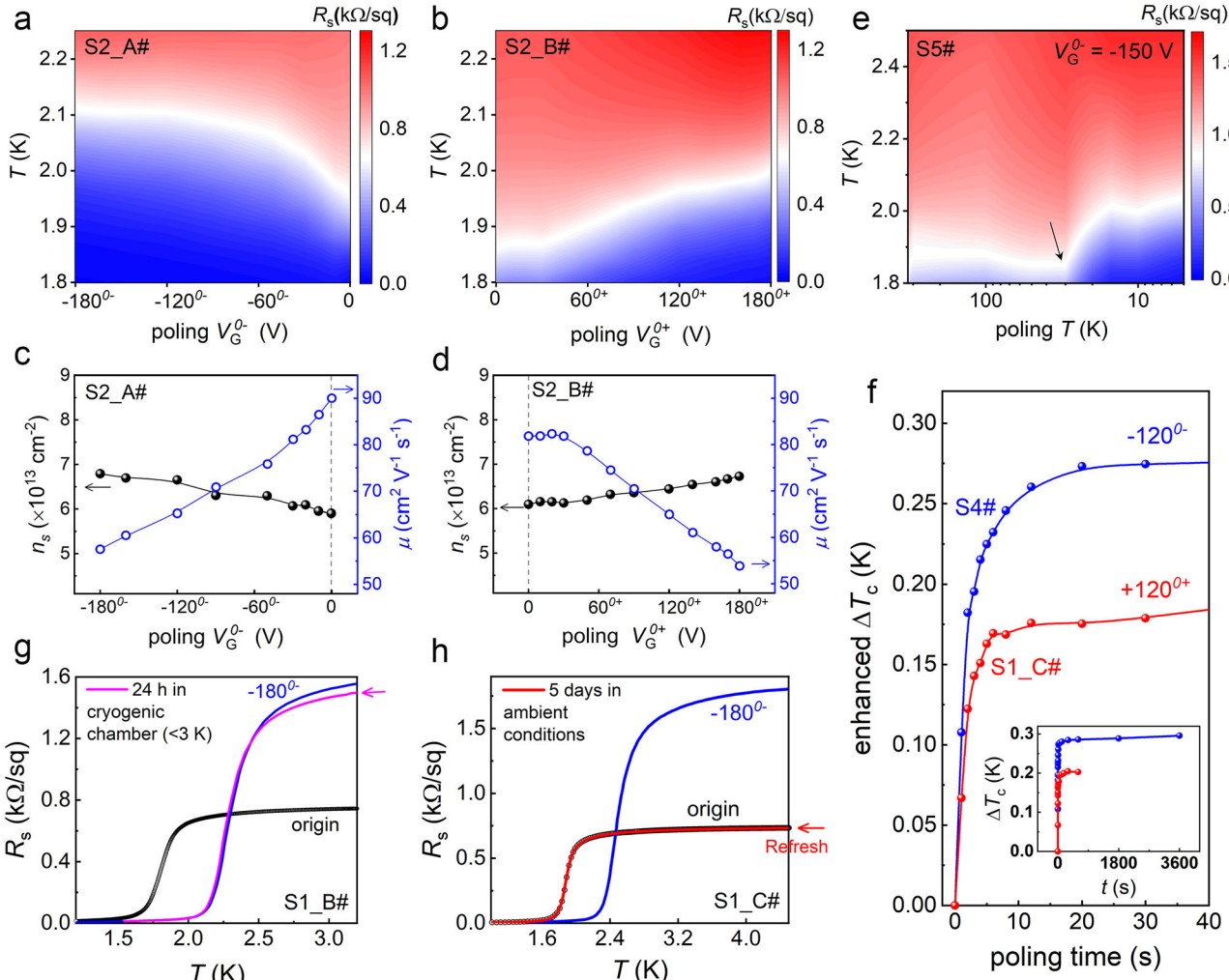

**Fig. 3 | Effects of poling $V_G$, temperature, time, and stability. a, b** Temperature-dependent $R_s(T)$ maps as a function of negative ($V_G < 0$) and positive ($V_G > 0$) poling $V_G$, respectively. **c, d** Corresponding carrier density ($n_s$) and mobility ($\mu$) measured at $T = 4.5$ K. **e** Temperature-dependent $R_s(T)$ map as a function of poling temperature, using a fixed poling $V_G = -150$ V. **f** Evolution of the enhanced $\Delta T_c$ with cumulative poling time, performed at $T = 4.5$ K with poling $V_G = \pm 120$ V on two different samples. The inset shows the same data on an extended time scale. **g** The poling-induced state remains non-volatile at low temperatures. **h** The poling-induced state recovers to the "origin" state after being left in ambient conditions for several days. For each experiment, fresh LAO/KTO samples were used as needed to ensure measurements began from the "origin" state.

its normal-state value at 4.5 K), was observed. Figures 2b, c display the $R_s(T)$ curves during a cycle where $V_G$ was swept from +180 V to −180V and then back. In both sweeping directions, a clear overall tuning effect consistent with previous studies[13,38] was observed: a positive (negative) $V_G$ decreases (increases) the normal-state $R_s$ and lowers (raises) $T_c$. However, in addition to this overall tuning effect, the $R_s(T)$ curves exhibit a strong dependence on the gating history. For example, the three different $V_G = 0$ states ("origin", "$O^+$", and "$O^-$", where "$O^+$" and "$O^-$" represent the $V_G = 0$ state after removing positive or negative $V_G$, respectively) yielded significantly different $R_s(T)$ curves.

As summarized in Fig. 2d−f, two notable and correlated features emerge beyond the overall tuning effect. First, during the initial run starting from the "origin" state (indicated by the blue lines), the device undergoes an irreversible change likely associated with polarization formation, corresponding to the universal $T_c$ enhancement. Second, after the initial run, the device exhibits repeatable and pronounced hysteresis in both $V_G$-$R_s$ (Fig. 2d) and $V_G$-$T_c$ (Fig. 2e) loops. Analysis of the $V_G$-$R_s$ loops at different temperatures, measured during both decreasing (Fig. 2g) and increasing (Fig. 2h) temperature orders, reveals that hysteresis begins above 50 K and becomes pronounced below 30 K. This temperature-dependent behavior closely matches

previous observations[39–41] of weak spontaneous polarization (0.04 $\mu$C/cm$^{-2}$) in KTO, where defect-induced polar nanoregions develop macroscopic coherence within this specific temperature range.

## Effects of poling magnitude, temperature, and time

Further investigations of the $O^\pm$ states after poling with $V_G$ (denoted as $V_G^{0\pm}$) were conducted to gain deeper insights into the observed ferroelectric behaviors. Since LAO/KTO is sensitive to gating history, we ensured an "origin" state in different experiments by using fresh samples when necessary. Figure 3a−d show the effects of poling $V_G$ polarity and magnitude. Two identical Hall bar devices (S2_A# and S2_B#) cut from the same LAO/KTO sample were used. Similar experiments were also conducted on a single device (S7#, Supplementary Fig. 1), whose "origin" state was regenerated through a refreshing process (described below).

For each polarity, the $V_G$ was gradually increased from 0 to $\pm 180$ V. At each $V_G$, we applied the poling $V_G$ at 4.5 K for 3 min, followed by $R_s(T)$ and Hall effect measurements after removing $V_G$ to 0. As shown in Fig. 3a, b, and Supplementary Fig. 1a, the $T_c$ increases with the magnitude of poling $|V_G|$ for both polarities, and $T_c$ for "$O^-$" is consistently higher than for "$O^+$". Poling up to $\pm 180$ V, corresponding

to ± 3.6 kV/cm, did not saturate $T_c$ enhancement, suggesting that the ferroelectric polarization remains unsaturated. As shown in Fig. 3c, d, and Supplementary Fig. 1b, the increase in $T_c$ was accompanied by a clear decrease in mobility $\mu$ (with a slight increase in carrier density $n_s$; however, the change in $n_s$ was much smaller than that of $\mu$). This indicates that, as discussed below, the ferroelectric polarization primarily modulates the interfacial potential well rather than directly altering $n_s$. It should be noted that the transport coefficients, $n_s$ and $\mu$, derived from Hall measurements on Hall bar devices, represent spatially weighted averages of all contributing layers, as these parameters are inherently depth-dependent along the $z$-direction.

Poling temperature also plays a critical role. As shown in Fig. 3e, $T_c$ begins to increase sharply when poling temperatures dropped below 30 K, matching the temperature at which $V_G$-$R_s$ hysteresis becomes pronounced (Fig. 2g, h). Notably, significant $T_c$ enhancement was observed after just one sec of poling, with saturation occurring after 10–15 sec of cumulative poling time (Fig. 3f and Supplementary Fig. 2). The tuned states remain nonvolatile at low temperatures (Fig. 3g). Full recovery to the "origin" state can be achieved by leaving the samples at ambient conditions for several days (Fig. 3h and Supplementary Fig. 3).

### Induced ferroelectricity in KTO

All these experimental observations support that ferroelectricity emerges in the KTO side of LAO/KTO interfaces. This is unsurprising, as ferroelectricity has previously been suggested in KTO through minimal chemical doping (e.g., Nb, Li)[42], oxygen deficiency[43], strain[44], or randomly distributed defects[40,45,46]. Notably, even nominally pure KTO samples typically contain substantial defect concentrations (~$10^{17}$ cm³). These defects promote the formation of polar nanoregions, which collectively drive KTO into a weak ferroelectric state at low temperatures[40,46]. Further evidence for low-temperature ferroelectricity in LAO/KTO comes from our Raman scattering measurements (Supplementary Note 1 and Supplementary Fig. 4), where the $TO_2$ and $TO_4$ optical modes emerge below ~40 K (~60 K in poled samples) – closely coinciding with the onset of hysteresis in the $V_G$-$R_s$ loops (Fig. 2g, h). The similarities between the observed hysteresis and Raman features in LAO/KTO and those in the intentionally ferroelectric Al/Sr$_{0.99}$Ca$_{0.01}$TiO$_3$ interface[29] provide additional support for ferroelectricity in LAO/KTO. We attribute the induced ferroelectricity to a field-induced ordering of the polar nanoregions that are present in KTO, which are inherently disordered and lacking long-range correlation in the pristine state.

While $V_G$-induced irreversibility in $R_s$ has been previously observed in SrTiO$_3$-based heterostructures and attributed to charge trapping/detrapping effects[47–49], the behavior we observe in LAO/KTO is fundamentally distinct. Our system exhibits reproducible and switchable hysteresis that is independent of the initial $V_G$ sweep direction–a hallmark of ferroelectricity (see Supplementary Fig. 5). Moreover, the limited magnitude of carrier density modulation in LAO/KTO (Supplementary Fig. 6) is incompatible with charge trapping, which would require significantly larger $n_s$ variations[47,48]. Although ionic migration could theoretically produce switchable polarization, this process is typically slow and thermally activated[50–52], conflicting with our low-temperature results. Taken together, the observed hysteresis and switching behaviors in LAO/KTO cannot be explained by charge trapping or ionic migration, providing compelling evidence for ferroelectricity as the governing mechanism.

### Coexistence of ferroelectricity and superconductivity in the conducting layer

As illustrated in Fig. 1a, superconductivity at KTO interfaces is confined within a thin KTO layer (~5–10 nm thick[11–13,38]), corresponding to the width of the interfacial potential well. The remaining KTO bulk serves as a thick insulator across which $V_G$ is applied. While ferroelectric polarization is expected in the insulating bulk, we propose that it also

occurs in the conducting layer. If polarization were confined solely to the insulating bulk, as in conventional ferroelectric transistors, its effects would primarily manifest in $n_s$ rather than $\mu$, inconsistent with our observations. Moreover, the bistability in transport properties suggests a switchable polarization within the conducting layer, whose built-in field modulates the interfacial potential well. These results lead us to conclude that ferroelectric polarization coexists with 2D superconductivity in the interfacial conducting layer. Although ferroelectricity and conductivity are traditionally considered mutually exclusive, polarization in the conducting KTO layer could emerge via coupling with the ferroelectric KTO bulk, mediated by their shared lattice bonding, while potentially developing distinct interfacial behavior.

### Ferroelectric polarization modulating the interfacial potential well

We now address how the presence of ferroelectricity in the conducting KTO layer can explain our experimental observations. Before delving into the specific phenomena associated with ferroelectricity, we reinforce that the transport properties of LAO/KTO are largely governed by the interfacial potential well profile, with $\mu$ serving as a key indicator. Previous studies[13,36,38] have shown that, particularly when $n_s$ is relatively large, gating at KTO interfaces primarily modulates $\mu$ rather than $n_s$. As shown in Fig. 2f and Supplementary Fig. 6a, sweeping $V_G$ from −180 V to 180 V caused $\mu$ to vary from ~20 cm² V⁻¹ s⁻¹ to ~90 cm² V⁻¹ s⁻¹, while $n_s$ changed only slightly (and in this device, in a manner opposite to that expected from a simple capacitance effect. See Supplementary Figs. 6b-d for more information). This behavior can be attributed to $V_G$-induced modulation of the interfacial potential well, which alters the spatial distribution of carriers[13,53,54]. Such modulation influences the "effective disorder", thereby affecting $\mu$ (a narrower potential well reduces $\mu$)[13,53,54].

Notably, the hysteresis observed in $V_G$-$R_s$ (Fig. 2d) and $V_G$-$T_c$ (Fig. 2e) loops is mirrored in $V_G$-$\mu$ loops (Fig. 2f) but not in $V_G$-$n_s$ loops (Supplementary Fig. 6a). Furthermore, the universal $T_c$ enhancement is accompanied by a significant decrease in $\mu$ (Figs. 2f, 3c, d, and Supplementary Fig. 1b), which can be attributed to narrowing of the interfacial potential well[13,36,38]. Therefore, in LAO/KTO interfaces, unlike typical ferroelectric effects, ferroelectricity primarily affects transport by modulating the interfacial potential well profile rather than directly altering $n_s$. We note that the enhancement of $T_c$ with potential well narrowing is a well-established experimental phenomenon in KTO-based interfaces[13,38], yet the underlying mechanism remains elusive. Several potential mechanisms may be at play: (i) fractal superconductivity induced by strong scattering[55], (ii) enhanced spin-orbit coupling energy[38], (iii) increased local three-dimensional carrier density, and (iv) modifications to electron-phonon coupling[56]. Further studies are needed to discern which of these mechanisms, if any, contribute to the observed enhancement.

Ferroelectric polarization in the interfacial KTO layer has two key effects: (1) it reduces the dielectric constant ($\varepsilon$) (Fig. 4a) and (2) introduces a built-in electric field due to polarization and the associated screening charges (insets of Fig. 4b). KTO, as an quantum paraelectric, exhibits a large $\varepsilon$ of up to 4500 at low temperatures[17]. However, $\varepsilon$ decreases dramatically under applied fields[57–59] and induced polarization[39,41]. As illustrated in Fig. 4a, after poling, ferroelectric polarization, regardless of its direction, lowers $\varepsilon$[39,41], thereby narrowing potential well (solid blue and red lines vs. dashed line, Fig. 4b), which explains the universal $T_c$ enhancement. Bistability naturally arises from switchable polarization, which can either narrow or widen the potential well depending on its direction. As shown in Fig. 4b, the built-in electric field ($E_{built-in}$) forming the potential well results from the superposition of two competing contributions: $E_P$ from ferroelectric polarization (opposing the gate field direction) and $E_S$ from surface screening charges. After $V_G$ removal, the persistent

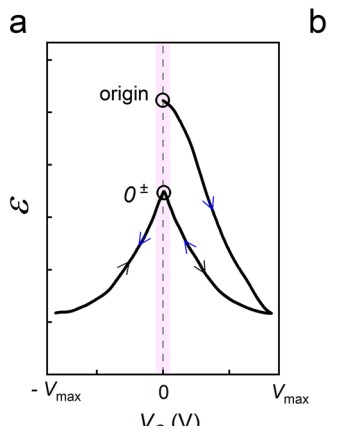

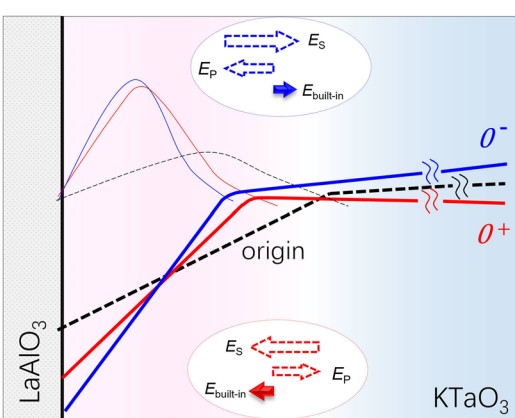

**Fig. 4 | Effect of ferroelectric polarization on the interfacial potential well.**
**a** Schematic evolution of the dielectric constant ($\varepsilon$) of KTO under an applied external electric field, reproduced based on experimental results from refs. 39,41. The pink shading highlights the "$O^{\pm}$" states, which exhibit a reduced $\varepsilon$ compared to the "origin" state. **b** Potential well modulation (thick lines) and electron envelope wavefunction (thin lines) for original (dashed), "$O^{-}$" (blue), and "$O^{+}$" (red) states. Insets show competing fields from $E_P$ (ferroelectric polarization) and $E_S$ (screening charges). The net field $E_{\text{built-in}} = E_P + E_S$ modulates the well profiles. Post $V_G$ removal, persistent tuning (aligned with $V_G$ direction) suggests that $|E_S| > |E_P|$.

residual field–dominated by $E_S$ which opposes and exceeds $E_P$–creates two distinct states: (1) In the "$O^{-}$" state, the net field $E_{\text{built-in}} = E_P + E_S$ narrows the potential well, enhancing carrier confinement and increasing $T_c$; (2) conversely, in the "$O^{+}$" state, the net field broadens the well, reducing $T_c$.

As shown in the $V_G = 0$ region of Fig. 2e, the $T_c$ enhancement after initial run is over 3 times larger than the $T_c$ difference between the "$O^{+}$" and "$O^{-}$" states in subsequent runs. This indicates that the reduction in $\varepsilon$ due to polarization has a stronger influence on the potential well profile than built-in field of the polarization alone. The inability to fully recover the origin state by warming the samples above the ferro-electric onset temperature (even to room temperature; see Supple-mentary Fig. 3a) is likely due to unbalanced screening charges[60,61] at the insulating LAO surface persisting in the cryostat environment. Upon cooling, these charges act as a poling field, reinducing polarization.

Our work demonstrates the coexistence of superconductivity and ferroelectricity at LAO/KTO interfaces, offering a unique platform for studying their interplay. The switchable polarization modulates the interfacial potential well by reducing $\varepsilon$ and altering built-in field, driving $T_c$ enhancement and bistable transport properties. These findings deepen our understanding of the mechanisms coupling superconductivity and ferroelectricity, and open new possibilities for designing multifunctional quantum devices. Prospectively, the doping tunable ferroelectricity in KTO[42] may enable LAO/KTO hetero-structures to serve as a programmable quantum platform[62,63] for controlling superconducting-ferroelectric coupled states.

## Methods
### Sample fabrication
LAO/KTO interface samples were fabricated by depositing amorphous LAO films onto 0.5 mm thick KTO single-crystalline substrates using pulsed laser deposition (PLD). A 248-nm KrF excimer laser was employed with a laser fluence of 0.7 J/cm² and a repetition rate of 10 Hz. A single-crystalline LAO target was used. For typical samples, 7-20 nm thick LAO films were grown at 300 °C in an atmosphere of $1 \times 10^{-5}$ mbar $O_2$ and $1 \times 10^{-7}$ mbar water vapor[13]. After deposition, the samples were cooled to room temperature under the same atmo-spheric conditions. In addition, "non-typical" samples were prepared by depositing a 2 nm "typical" LAO layer, followed by a 20 nm LAO layer grown at room temperature. The LAO films are highly insulating, with conduction confined to a thin (~5–10 nm thick) KTO layer adjacent to the interface.

### Hall bar devices
Hall bar structures were patterned onto KTO substrates using standard optical lithography and lift-off techniques, with ~200 nm thick $AlO_x$ films serving as a hard mask[35]. The $AlO_x$ films were deposited by PLD at room temperature under base pressure, with a laser fluence of 2.5 J/cm². To ensure high insulation in the $AlO_x$-covered areas, the patterned substrates were annealed at 300 °C for 2 h in a flow of 1 bar $O_2$.

Subsequently, LAO films were deposited onto these pre-patterned KTO substrates, forming conducting LAO/KTO interfaces exclusively in the uncovered regions. Each sample contained four identical Hall bar devices. Across different samples, the central Hall bar bridges were fabricated in two sizes: 20 μm in width and 100 μm in length, or 100 μm in width and 500 μm in length.

### Electrical contact
Electrical contacts to the conducting LAO/KTO interfaces were estab-lished using ultrasonic bonding with Al wires.

### Gating and poling
As illustrated in Fig. 1a, a back-gating voltage ($V_G$) was applied between the conducting interface and the bottom silver electrode, with the polarity defined relative to the interface. The value of $V_G$ represents the bias applied to the bottom silver electrode. Throughout the gating process, the leakage current was consistently below 10 nA.

The process of applying $V_G$ and subsequently setting it to 0 is referred to as "poling", similar to operations in conventional ferro-electrics. The states "$O^{+}$" and "$O^{-}$" represent the $V_G = 0$ state after removing positive and negative $V_G$, respectively. For clarity, we also denote the $V_G = 0$ state after poling with $V_G$ as $V_G^{0\pm}$. For example, "$120^{0+}$" indicates the $V_G = 0$ state following an applied $V_G = +120$ V. This nomenclature is extended to all cases accordingly.

### Transport measurements
Low-temperature transport measurements were performed using a commercial ⁴He cryostat equipped with a ³He insert (Cryogenic Ltd.). A four-probe DC technique was employed, utilizing a Keithley 6221 current source and a Keithley 2182 A nanovoltmeter. Carrier mobility ($\mu$) and density ($n_s$) were obtained from Hall effect measurements conducted on Hall bar devices.

### Raman spectroscopy
The low-temperature Raman measurements were performed in a cryostat (Attocube, attoDRY2100) equipped with a commercial

confocal microscope (Horiba, LabRam Odyssey). The sample temperature was controlled in the range of 1.6 K to 300 K. A linearly polarized 532 nm laser was used as the excitation source, with a ~1 μm beam radius and ~3.6 mW power at the sample position.

## Data availability

All data that support the key findings in this study are available within the main text and Supplementary Information. Additional raw data can be obtained from the corresponding authors upon request. Source data are provided with this paper.

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

## Acknowledgements

This work was supported by the National Key R&D Program of China (Grant No. 2023YFA1406400 to Y.X.), National Natural Science Foundation of China (Grant No. 12325402 and 12534005 to Y.X., Grant No. 12504226 to M.Z.), China Postdoctoral Science Foundation (Grant No. 2025M773422 to M.Z.) and Innovation Program for Quantum Science and Technology (Grant No.2021ZD0300200 to Y.X.).

## Author contributions

Y.X. and M.Z. conceived the study and proposed the strategy. M.Z., M.Q. and Y.S. prepared the samples. S.H. contributed to the development of the electrostatic measurements. M.Z., M.Q. and Y.S. conducted transport measurements and performed analysis. Y.Z. contributed to the interpretation of the physical mechanisms. All authors participated in the discussion on the paper. M.Z. and Y.X. wrote the manuscript with input from all the authors.

## Competing interests

The authors declare no competing interests.
