## [Transparent Peer Review file · Nature Communications]

Enhanced superconductivity and coexisting ferroelectricity at oxide interfaces

Corresponding Author: Professor Yanwu Xie

Version 0:

Reviewer comments:

Reviewer #1

(Remarks to the Author)

This manuscript by Zhang et al. delves into two unusual phenomena occurring during the gate-voltage tuning process in the LaAlO₃/KTaO₃(111) (LAO/KTO) heterostructure: (i) After cycles of gating and poling, the superconducting transition temperature T_c constantly becomes higher than the original value. (ii) The normal-state resistance changes accordingly in a bistable manner that relies on the polarity of voltage. The authors attribute these observations to the gate-voltage manipulated ferroelectricity of KTO near the interface, which determines the width of electron wavefunction distribution for the interfacial 2D electron gas (2DEG).

I found this manuscript to be of sufficiently high quality. The analysis and discussion are well-reasoned, supplied by an impressive amount of experimental data. Nonetheless, I hope the author can provide some reassurances about the general interest of these results. Otherwise, I still hesitate to recommend publication.

It has been repeatedly stressed that the controllable ferroelectric/ferroelastic properties of SrTiO₃ (STO) [e.g., B. Kalisky, et al. Nat. Mater. 12, 1091 (2013); C. Cen, et al. Nat. Mater. 7, 298 (2008); Y.-Y. Pai, et al. Phys. Rev. Lett. 120, 147001 (2018); M. Yu, et al. Phys. Rev. X 15, 011037 (2025)] and KTO [M. Yu, et al., Nano Lett. 22, 6062-6068 (2022)] have significant impact on the behavior of interfacial 2DEG, especially the superconducting pairing. Likewise, it is also well known that the interfacial superconductivity can be effectively tuned by changing the width of interfacial potential well (Refs. [13], [48]). The present work establishes a link between these two understandings. Such a link is, however, not very surprising or unexpected; at this stage, I am not sure if it represents a substantial achievement in this field.

There are several further questions that probably lead to new insights. First, I wonder what is the underlying mechanism for the dependence of superconducting properties (T_c , superfluid density, etc) on the width of interfacial potential well (or depth of the electron wavefunction's extension from interface to the KTO side). Why the superconductivity changes substantially --- becomes more "unconventional" with an increase of T_c --- upon squeezing the itinerant electrons towards the interface? Since the width is supposed to be a few nanometers, it cannot be a simple dimension effect. Then what indeed matters? Is it the spin-orbit coupling effect at the interface that serves as the crucial factor? Clarification of this point is of primary importance.

My second question concerns the statement that ferroelectric polarization and metallic conductivity (as well as superconductivity) coexist in the top layers of KTO in proximity to the interface. It undoubtedly means that the carrier density is not uniform in the conducting layer, but develops a gradient along the normal direction of the interface. In this sense, how meaningful are the parameters such as Hall carrier density n_H and Hall mobility μ_H ? Indeed, these parameters can be functions of the distance z from the interface and are thereby spatially varying; the high mobility can potentially stem from the carriers farther from the interfacial scattering centers, yet this does not mean the absence of slower electrons in the vicinity of interface (they are simply short-circuited by their faster counterparts in deeper positions). Consequently, the interfacial superconductivity may also be nonuniform and z -dependent? What is the best picture describing the superconducting system in the oxide heterointerfaces?

I am looking forward to seeing the authors' reply to my questions and my final evaluation would be made based on those responses.

A relatively minor point: for the difference between 0+ and 0- states, seems the interpretations are mainly presented in the caption of Fig. 4b; the main text contains very limited contents of corresponding discussion. Probably this can be balanced to some extent.

Reviewer #2

(Remarks to the Author)

The manuscript claimed about the coexistence of superconductivity and ferroelectricity in KTaO_3 (111) based heterostructure. The authors further attribute the enhancement of superconducting T_c to the ferroelectric polarization induced reduction in dielectric constants. While the authors have demonstrated superconductivity, there is no direct experimental evidence about ferroelectricity. All claims about ferroelectricity are rather speculative, which may be true or may not be true as alternative scenario can also be attributed to the experimental claims.

Another major issue I found that the authors have completely ignored the issue of charge trapping/detrapping under electrical gating. This has been demonstrated extensively in context of SrTiO_3 based heterostructures [Sci. Rep. 4, 6788 (2014); Phys. Rev. Lett. 124, 017702 (2020); Phys. Rev. Applied 15, 054008 (2021)]. Such charge trapping phenomena also gives rise to hysteresis in resistance vs. gate voltage measurement. Similar observations have been reported in Fig. 2 of this manuscript but attributed to the ferroelectricity. The authors need to demonstrate that their observations are not linked with charge trapping phenomena.

Because of these two significant issues in the manuscript, I do not recommend the publication of this paper in Nature Communications.

Version 1:

Reviewer comments:

Reviewer #1

(Remarks to the Author)

I have read the authors' response and the revised manuscript. I am impressed by the scientific care and scrutiny the authors show in addressing all concerns and questions raised in the first round of review. In general, I am satisfied with the responses and revisions. The manuscript has been extensively improved, particularly with the scientific significance clearly stressed and validity of main conclusions strengthened by the newly-supplied Raman data. I also appreciate that the authors mentioned Feigel'man's fractal superconductivity as a potential origin of T_c enhancement. Taking all these together, my recommendation would be to publish this work in Nature Communications.

Meanwhile, I have one remaining suggestion. As elucidated by the authors in the reply file, the transport coefficients derived from the Hall measurement, including n_s and μ , are weighted spatial average of all contributing layers since these parameters are inherently dependent on the depth along z direction. This point is not mentioned in the manuscript, while I think it is helpful to the reader to see it addressed. Probably a brief note at the end of the paragraph introducing these parameters (the first paragraph on page 5) can resolve this matter. Also, n_s and μ are only defined in the Methods but not in the main text.

Reviewer #2

(Remarks to the Author)

I have gone through the revised manuscript along with the response letter and supplemental materials. I am still not convinced by the claim about the ferroelectricity. The authors have performed Raman spectroscopy, and the polar mode TO2 and TO4 are present in every sample they examined, including a bare KTO (111) substrate. While the mode confirms the presence of polar distortion, it does not confirm ferroelectricity. The polar distortion may not be uniform and can be confined within a few unit cells. It is well known that depending on the concentration of polar nano regions and temperature, one can get several phases, including paraelectric, ferroelectric, and dipolar glass phase [see J. Phys. Condens. Matter 15, R367–R411(2003)]. In fact, in one recent article [Nature Communications 15, 3830 (2024)], it has been demonstrated the electron dynamics of metallic KTaO_{3-x} sample behaves like a glass, which is related to the dipolar glassy phase of polar nano regions. Overall, the experimental results reported in this paper do not confirm ferroelectricity, although that may be a possibility. So, I do not recommend the publication of this manuscript in Nature Communications in its present form.

Manuscript: NCOMMS-25-06326-T

Title: “Universally enhanced superconductivity and coexisting ferroelectricity at oxide interfaces”

Dear Reviewers,

Thank you for your thoughtful feedback on our manuscript. Your comments have been invaluable in strengthening this work. We have carefully addressed all points through additional experiments, enhanced discussions, and textual revisions.

This response letter is organized as follows:

- **Part A:** Summary of major revisions
- **Part B:** Point-by-point responses to reviewer comments

All changes are highlighted in the revised manuscript for ease of review. We appreciate the opportunity to improve this work and are happy to provide further clarification if needed.

Sincerely,

Yanwu Xie and Meng Zhang

On behalf of all the authors

Part A: List of major changes (Colored in red in the revised manuscript)

1. Divided the original manuscript into the main text and Supplementary Information.
2. Added Prof. Yi Zhou (theoretical physicist) to the author list for his substantial contribution in revising the manuscript and addressing reviewer comments, particularly regarding the mechanism of T_c enhancement with the potential well narrowing.
3. Revised the Introduction and Conclusion sections to better highlight the significance of this work.
4. Added Raman spectroscopy measurements (Supplementary Note 1 and Supplementary Fig. 4), with corresponding revisions to the main text and Methods section.
5. Performed additional experiments tracking the evolution of R_s with V_G under continuous gate voltage sweeps (shown in Supplementary Fig. 5).
6. Substantially revised the “Induced ferroelectricity in KTO” section (Pages 5&6) to strengthen evidence for ferroelectricity while explicitly ruling out charge trapping as the primary mechanism.
7. Revised the concluding sentence of the “Coexistence of ferroelectricity and superconductivity in the conducting layer” section (Page 6) to clarify the relationship between interfacial and bulk ferroelectricity.
8. Added a discussion paragraph (Page 7) exploring potential mechanisms underlying T_c enhancement induced by potential well narrowing.
9. Reallocated the $0^+/0^-$ state difference interpretation between the main text and Fig. 4b legend.
10. Updated references, with the following additions (in the main text):

- [1] Kalisky, B. et al. *Nat. Mater.* **12**, 1091-1095 (2013)
- [2] Cen, C. et al. *Nat. Mater.* **7**, 298–302 (2008)
- [3] Pai, Y.-Y. et al. *Phys. Rev. Lett.* **120**, 147001 (2018)
- [4] Biscaras, J. et al. *Sci. Rep.* **4**, 6788 (2014)
- [5] Yin, C. et al. *Phys. Rev. Lett.* **124**, 017702 (2020)
- [6] Ojha, S. K. et al. *Phys. Rev. Appl.* **15**, 054008 (2021)
- [7] Feigel'man, M. V. et al. *Ann. Phys.* **325**, 1390–1478 (2010)
- [8] Chen, X. et al. *Nat. Commun.* **15**, 7704 (2024)
- [9] Yu, M. et al. *Nano Lett.* **22**, 6062–6068 (2022)
- [10] Yu, M. et al. *Phys. Rev. X* **15**, 011037 (2025)

Part B-1: Responses to the comments from Reviewer #1

This manuscript by Zhang et al. delves into two unusual phenomena occurring during the gate-voltage tuning process in the $\text{LaAlO}_3/\text{KTaO}_3(111)$ (LAO/KTO) heterostructure: (i) After cycles of gating and poling, the superconducting transition temperature T_c constantly becomes higher than the original value. (ii) The normal-state resistance changes accordingly in a bistable manner that relies on the polarity of voltage. The authors attribute these observations to the gate-voltage manipulated ferroelectricity of KTO near the interface, which determines the width of electron wavefunction distribution for the interfacial 2D electron gas (2DEG).

Reply: We sincerely thank the reviewer for their careful reading and insightful summary of our work.

I found this manuscript to be of sufficiently high quality. The analysis and discussion are well-reasoned, supplied by an impressive amount of experimental data. Nonetheless, I hope the author can provide some reassurances about the general interest of these results. Otherwise, I still hesitate to recommend publication.

Reply: We sincerely thank the reviewer for their positive assessment of our manuscript's quality and experimental rigor. Below we address the broader significance of these findings (additional details are provided in our response to the reviewer's subsequent comment):

(1) Scientific novelty:

Our work reports two unprecedented phenomena in oxide interfaces:

- First demonstration of electric-field-controlled **non-volatile T_c enhancement**
- First observation of **polarity-switchable** bistable states with coupled superconducting properties.

(2) Potential impact:

(i) Fundamental advance

While superconductor-ferroelectric coexistence is uncommon, **electrically switchable** systems are **exceptionally rare** (the only comparable reports involve non-oxide systems, Ref. [6]). Our findings establish LAO/KTO as a promising platform for studying intrinsic superconductor-ferroelectric coupled states.

(ii) Tunability prospects:

Beyond impurity-induced ferroelectricity, KTO exhibits doping-tunable polarization (Ref. [42]), suggesting that the superconducting-ferroelectric coupling in LAO/KTO heterostructures can be systematically controlled.

Revision summary: In the revised manuscript, we have substantially strengthened both the introduction and conclusion sections to better articulate these scientific implications.

Introduction (revised, Page 2):

“In STO-based interfaces, superconducting-ferroelectric coexistence has been achieved via ^{18}O substitution¹⁹, and ferroelectric-switchable 2D electron gases have been realized through Ca alloying²⁹. Furthermore, the effects of ferroelectric/ferroelastic domains on both conductivity^{30,31} and superconducting pairing³² have been suggested. However, the full integration of superconductivity with electrically switchable ferroelectric polarization, despite being conceptually proposed²⁹, has remained experimentally unrealized.”

Conclusion (added, Page 9):

“Prospectively, the doping tunable ferroelectricity in KTO⁴² may enable LAO/KTO heterostructures to serve as a programmable quantum platform^{62,63} for controlling superconducting-ferroelectric coupled states.”

It has been repeatedly stressed that the controllable ferroelectric/ferroelastic properties of SrTiO₃ (STO) [e.g., B. Kalisky, et al. Nat. Mater. 12, 1091 (2013); C. Cen, et al. Nat. Mater. 7, 298 (2008); Y.-Y. Pai, et al. Phys. Rev. Lett. 120, 147001 (2018); M. Yu, et al. Phys. Rev. X 15, 011037 (2025)] and KTO [M. Yu, et al., Nano Lett. 22, 6062-6068 (2022)] have significant impact on the behavior of interfacial 2DEG, especially the superconducting pairing. Likewise, it is also well known that the interfacial superconductivity can be effectively tuned by changing the width of interfacial potential well (Refs. [13], [48]). The present work establishes a link between these two understandings. Such a link is, however, not very surprising or unexpected; at this stage, I am not sure if it represents a substantial achievement in this field.

Reply: We thank the reviewer for recognizing how our work bridges the understanding of ferroelectricity and superconductivity in oxide interfaces. Below we further clarify its significance:

(1) First demonstration of ferroelectric-switched superconductivity

The quantum paraelectric nature of STO and KTO indeed suggests ferroelectric fluctuations as a mechanism for superconducting pairing, making their coexistence an important research focus. As the reviewer pointed out, previous studies have suggested that the tunable ferroelectric/ferroelastic domains in STO may influence interfacial conductive behavior, potentially offering insights into superconducting pairing mechanisms. However, the experimental realization of *electrically controlled* superconducting states via ferroelectric switching has remained elusive due to two critical gaps:

- (i) **Coexistence without control:** While STO-based systems have shown ferroelectricity/superconductivity coexistence (Refs. [19,28]), the essential functionality of electrically switching superconductivity via ferroelectric polarization has never been achieved.

- (ii) **Switching without superconductivity:** While STO interfaces have demonstrated ferroelectric-switchable conduction (Ref. [29]; Noël *et al.*, *Nature* **580**, 483–486 (2020)), these studies either (a) did not achieve superconducting states or (b) did not explore the superconducting regime.

Our work bridges these gaps by providing **the first experimental demonstration of polarity-switchable superconductivity** in oxide interfaces.

(2) Further justification of significance

Achieving ferroelectric-controlled superconductivity is extremely challenging and requires overcoming non-trivial material constraints:

- (i) **STO's inherent limitations:** The remanent polarization ($\sim 1 \mu\text{C}/\text{cm}^2 = 6 \times 10^{12} \text{ cm}^{-2}$) (Ref. [29]; Noël *et al.*, *Nature* **580**, 483-486 (2020)) is significantly smaller than the required superconducting carrier density. This mismatch, combined with STO's low T_c , makes switchable control very challenging. Thus, while previous studies in STO-based interfaces have revealed key aspects of ferroelectricity and superconductivity, achieving ferroelectric-controlled superconductivity in STO-based interfaces remains elusive.
- (ii) **KTO's advantages.** In contrast, KTO interfaces exhibit both higher T_c and a distinct tuning mechanism primarily governed by interfacial potential modulation rather than carrier density. These intrinsic properties enable the switchable ferroelectric control reported here.
- (iii) **Wider significance:** Across all known condensed matter systems, electrically switchable ferroelectric control of superconductivity remains exceptionally rare - highlighting both the fundamental challenge and the importance of our results.

Revision summary: To address the reviewer's comment, we have modified the introduction in the revised manuscript (Page 2) as follows:

“In STO-based interfaces, superconducting-ferroelectric coexistence has been achieved via ^{18}O substitution¹⁹, and ferroelectric-switchable 2D electron gases have been realized through Ca alloying²⁹. Furthermore, the effects of ferroelectric/ferroelastic domains on both conductivity^{30,31} and superconducting pairing³² have been suggested. However, the full integration of superconductivity with electrically switchable ferroelectric polarization, despite being conceptually proposed²⁹, has remained experimentally unrealized.”

There are several further questions that probably lead to new insights. First, I wonder what is the underlying mechanism for the dependence of superconducting properties (T_c , superfluid density, etc) on the width of interfacial potential well (or depth of the electron wavefunction's extension from interface to the KTO side). Why the superconductivity changes substantially --- becomes more “unconventional” with an increase of T_c --- upon squeezing the itinerant electrons towards the interface? Since the width is supposed to be a few nanometers, it cannot be a simple dimension effect.

Then what indeed matters? Is it the spin-orbit coupling effect at the interface that serves as the crucial factor? Clarification of this point is of primary importance.

Reply: We thank the reviewer for raising this critical question regarding the mechanism linking interfacial potential well width (i.e., carrier confinement) to unconventional superconductivity enhancement in LAO/KTO interfaces. While it is still an open question in the field, our manuscript attributes the phenomenon primarily to **ferroelectricity-induced reduction in dielectric constant**, which modulates the potential well profile and carrier distribution. Below we clarify the key points raised:

(1) Mechanism for T_c enhancement and unconventionality:

- (i) As detailed in Sections "Ferroelectric polarization modulating the interfacial potential well" (Page 7) and Fig. 4, ferroelectric polarization in the interfacial KTO layer *reduces the dielectric constant* (ϵ). This reduction occurs because polarization suppresses quantum paraelectric fluctuations inherent to bulk KTO (Refs. [39, 41, 57-59]).
- (ii) A lower ϵ directly *narrows the interfacial potential well* (Fig. 4b), confining the electron carriers closer to the interface. Crucially, this confinement *increases carrier scattering* (evidenced by decreased mobility μ , Figs. 2f, 3c-d, Supplementary Fig. 1b) but *paradoxically enhances T_c* .
- (iii) This counterintuitive result — where increased disorder (lower μ) correlates with *higher T_c* — signals a departure from conventional disorder-sensitive superconductivity.
- (iv) This seeming paradox can be resolved via the fractal superconductivity scenario: In disordered systems, fractal geometry can enhance superconducting T_c by (a) creating localized superconducting "puddles" with higher local T_c , and (b) enabling percolative phase coherence. [See the famous paper: "Fractal superconductivity near localization threshold" M.V. Feigel'man, L.B. Ioffe, V.E. Kravtsov, and E. Cuevas, *Annals of Physics* 325 (2010) 1390-1478]
- (v) We note that the percolative nature of polar nanodomains in KTO (Refs. [40, 46]) may lead to the formation of fractal-like superconducting pathways.

(2) Role of potential well width vs. simple dimensionality:

- (i) The reviewer correctly notes that the well width (~5-10 nm) is too large for pure 2D quantum size effects to dominate T_c enhancement. Our data support that it is not the width *per se*, but the *change in the electrostatic environment* (specifically ϵ reduction) caused by ferroelectricity that is key.
- (ii) The evidence lies in the correlation between polarization-induced ϵ reduction, well narrowing, mobility decrease (indicating modified scattering), and T_c increase. Simple dimensional confinement without this dielectric change would likely suppress T_c via enhanced disorder without the compensating unconventional pairing mechanism.

(3) Alternative explanations for T_c enhancement:

While we ascribe the T_c enhancement primarily to ferroelectricity-induced dielectric constant reduction, we acknowledge several alternative mechanisms requiring further investigation.

- (i) **Spin-orbit coupling (SOC).** The reviewer correctly notes SOC's potential involvement. KTO's intrinsically strong bulk SOC likely contributes to the interfacial superconducting state, as shown by gate-modulation experiments [Hua et al., *npj Quantum Mater.* **7**, 97 (2022)] where suppression of the superconducting layer thickness occurs together with increases in both T_c and spin-orbit energy. However, while these phenomena are clearly correlated, we cannot establish a direct causal relationship between SOC enhancement and T_c elevation, leaving this as an open question.
- (ii) **Three-dimensional carrier concentration (n_{3d}) effects.** A straightforward consideration suggests that n_{3d} increases with potential well narrowing, which might elevate T_c .
- (iii) **Electron-phonon coupling modification.** Recent work identifies surface electron-phonon coupling as crucial for KTO superconductivity [Chen et al., *Nat. Commun.* **15**, 7704 (2024)]. This coupling may strengthen under progressive electron confinement near the interface.

Revision summary: To address the reviewer's question, in the revised manuscript on Page 7 we have added the following statement:

“We note that the enhancement of T_c with potential well narrowing is a well-established experimental phenomenon in KTO-based interfaces^{13,38}, yet the underlying mechanism remains elusive. Several potential mechanisms may be at play: (i) fractal superconductivity induced by strong scattering⁵⁵, (ii) enhanced spin-orbit coupling energy³⁸, (iii) increased local three-dimensional carrier density, and (iv) modifications to electron-phonon coupling⁵⁶. Further studies are needed to discern which of these mechanisms, if any, contribute to the observed enhancement.”

My second question concerns the statement that ferroelectric polarization and metallic conductivity (as well as superconductivity) coexist in the top layers of KTO in proximity to the interface. It undoubtedly means that the carrier density is not uniform in the conducting layer, but develops a gradient along the normal direction of the interface. In this sense, how meaningful are the parameters such as Hall carrier density n_s and Hall mobility μ_H ? Indeed, these parameters can be functions of the distance z from the interface and are thereby spatially varying; the high mobility can potentially stem from the carriers farther from the interfacial scattering centers, yet this does not mean the absence of slower electrons in the vicinity of interface (they are simply short-circuited by their faster counterparts in deeper positions). Consequently, the interfacial superconductivity may also be nonuniform and z -dependent? What is the best picture describing the superconducting system in the oxide heterointerfaces?

Reply: We sincerely appreciate the reviewer's insightful remark regarding this general question about oxide interfaces. We fully agree with the key points raised about the z -dependent nature of both transport parameters and superconductivity

(1) Regarding the parameters such as Hall carrier density and Hall mobility:

We note that this z -dependence is characteristic of nearly all heterointerfaces, including conventional semiconductor systems, when charge transfer occurs. The carrier distribution profiles can in principle be modeled via self-consistent Poisson-Schrödinger simulations. Importantly, despite this inherent z -dependence, Hall measurements remain the standard experimental technique for characterizing such heterointerfaces (including oxide interfaces), as they effectively capture the macroscopic transport behavior through weighted spatial averaging of all contributing layers.

(2) Concerning the z -dependence of superconductivity

We fully concur with the reviewer regarding the inherent depth dependence of interfacial superconductivity. While systematic z -resolved studies remain challenging for STO- and KTO-based interfaces (due to limitations in current experimental capabilities for growing atomically-resolved, high-quality doped STO and KTO films), compelling evidence from cuprate heterostructures **demonstrates clear z -dependence, with the overall superconducting properties being governed by the optimally superconducting sublayer.**

- (i) In model $\text{La}_2\text{CuO}_4/\text{La}_{1.55}\text{Sr}_{0.45}\text{CuO}_4$ bilayers studied via atomic-layer-resolved delta-doping, the optimal superconducting state occurs in the second CuO_2 plane from the interface [Logvenov et al., *Science* **326**, 699 (2009)].
- (ii) Our recent work (in review) demonstrates tunable depth profiles in $\text{La}_{2-x}\text{Sr}_x\text{CuO}_4/\text{LaSrCuO}_4$ bilayers, where the optimal superconducting CuO_2 plane systematically shifts from the 3rd ($x = 0$) to the 6th plane ($x = 0.15$) with Sr doping.

Thus, returning to STO- or KTO-based oxide interfaces, we speculate that—precisely as the reviewer suggests—the interfacial superconducting state exhibits z -dependence, with the measured properties dominated by the optimal superconducting sublayer.

I am looking forward to seeing the authors' reply to my questions and my final evaluation would be made based on those responses.

Reply: We sincerely appreciate the reviewer's thoughtful suggestions. We're pleased to confirm that all comments have been fully addressed through extensive responses and thorough manuscript revisions. These improvements now better reflect the significance of our findings. We'd be delighted to provide any additional information the reviewer may require.

A relatively minor point: for the difference between 0^+ and 0^- states, seems the interpretations are mainly presented in the caption of Fig. 4b; the main text contains

very limited contents of corresponding discussion. Probably this can be balanced to some extent.

Reply: We thank the reviewer for this constructive suggestion. The recommended modifications have been implemented accordingly in the revised manuscript.

Revision summary:

In the revised manuscript, we have added the following discussion regarding the “ O^- ” and “ O^+ ” states on Page 8:

“As shown Fig. 4b, the built-in electric field ($E_{\text{built-in}}$) forming the potential well results from the superposition of two competing contributions: E_P from ferroelectric polarization (opposing the gate field direction) and E_S from surface screening charges. After V_G removal, the persistent residual field—dominated by E_S which opposes and exceeds E_P —creates two distinct states: (1) In the “ O^- ” state, the net field $E_{\text{built-in}} = E_P + E_S$ narrows the potential well, enhancing carrier confinement and increasing T_c ; (2) conversely, in the “ O^+ ” state, the net field broadens the well, reducing T_c .”

We have also refined the caption of Fig. 4b for conciseness:

“**b**, Potential well modulation (thick lines) and electron envelope wavefunction (thin lines) for original (dashed), “ O^- ” (blue), and “ O^+ ” (red) states. Insets show competing fields from E_P (ferroelectric polarization) and E_S (screening charges). The net field $E_{\text{built-in}} = E_P + E_S$ modulates the well profiles. Post V_G removal, persistent tuning (aligned with V_G direction) suggests that $|E_S| > |E_P|$ ”

Part B-2: Responses to the comments from Reviewer #2

The manuscript claimed about the coexistence of superconductivity and ferroelectricity in KTaO_3 (111) based heterostructure. The authors further attribute the enhancement of superconducting T_c to the ferroelectric polarization induced reduction in dielectric constants. While the authors have demonstrated superconductivity, there is no direct experimental evidence about ferroelectricity. All claims about ferroelectricity are rather speculative, which may be true or may not be true as alternative scenario can also be attributed to the experimental claims.

Reply: We sincerely appreciate the reviewer's critical comment regarding the evidence of ferroelectricity in our system. In response, we have performed additional experiments (Raman scattering and additional transport) and strengthened our discussion to rigorously establish the ferroelectric origin of the observed phenomena. As detailed in our response to the reviewer's second comment, alternative interpretations like charge trapping/detrapping can be excluded. For clarity, we organize this reply into three sections (Section 2 contains the key experimental results).

(1) Established understanding of polar nanoregions in KTO

While KTO remains quantum paraelectric down to the lowest attainable temperatures in its pure form, this low-temperature paraelectric state is rather unstable. The ferroelectric phase can readily be induced through several well-documented approaches including (i) chemical doping, (ii) oxygen deficiency, (iii) applied strain, and (iv) naturally occurring defects (even nominally pure KTO samples contain substantial defect concentrations of $\sim 10^{17} \text{ cm}^{-3}$). These defects generate polar nanoregions (PNRs) that have been extensively characterized in previous studies (Refs. [40,45-46]). Importantly, as demonstrated in previous observations (Ref. [40]), these PNRs undergo cooperative ordering below a critical temperature, transforming KTO into a weak ferroelectric state with small but measurable spontaneous polarization ($\sim 0.04 \mu\text{C}/\text{cm}^2$) (Ref. [40]). The ferroelectric behavior we observed in our LAO/KTO heterostructures directly originates from this PNR-induced ferroelectricity that has been firmly established in the literatures (Refs. [40,45-46]).

(2) Raman spectroscopy evidence for polar nanoregions in LAO/KTO heterostructures

To directly demonstrate the existence of PNRs in our LAO/KTO heterostructures, we conducted Raman spectroscopy measurements—a well-established technique for detecting PNRs in KTO. For comprehensive comparison, we examined (i) as-grown LAO/KTO, (ii) poled LAO/KTO (following our defined protocol in the manuscript), (iii) KTO single-crystalline substrate, and (iv) commercial ferroelectric $\text{KTa}_{0.65}\text{Nb}_{0.35}\text{O}_3$ (KTN_{0.35}) single crystal. All spectroscopic data are presented in Figure R1 with detailed analysis in subsequent subsections.

Fig. R1 (also presented as **Supplementary Fig. 4** in the revised manuscript) | **Raman spectroscopy.** Temperature dependence of normalized Raman spectra of (a) ferroelectric KTN_{0.35}(111) single crystal, (b) KTO(111) substrate, and LAO/KTO(111) heterostructures in both (c) original and (d) \bar{O} state (poled at -180 V). Bold red curves highlight T^* (temperature where transverse polar optical phonon modes TO₂ and TO₄ emerge), with mode frequencies indicated by arrows. (e) Back-ground-subtracted TO₄ peak intensity versus temperature (inset shows saturation below 10 K). Comparison of original and \bar{O} state spectra at temperatures at (f) 60 K and (g) 1.6 K, with TO₄ regions magnified in insets.

(2.1) Background on KTO phonon modes

Three transverse polar optical phonon modes (TO₁, TO₂, and TO₄) are theoretically predicted and experimentally observed in perovskite KTO systems, including single crystals [H. Uwe et al. *Phys. Rev. B* 33, 6436-6440 (1986)] and ceramics [S. Glinsek et al. *J. Appl. Phys.* 111, 104101 (2012)]. The TO₁ mode corresponds to the relative vibration of Ta⁵⁺ ions against rigid O₆ octahedra (Slater mode), TO₂ reflects K⁺ ion motion against a rigid TaO₆ framework (Last mode), and TO₄ involves the bending of O₆ octahedra (Axe mode) [X. Wen et al. *Appl. Phys. Lett.* 122, 232905 (2023); S. Glinsek et al. *J. Appl. Phys.* 111, 104101 (2012); W. G. Nilsen et al. *J. Chem. Phys.* 47, 1413 (1967)]. While the soft TO₁ mode strongly softens upon cooling and exhibits low intensity, TO₂ and TO₄ show stable frequencies but gradually diminishing intensities with increasing temperature. Notably, these **Raman-inactive** modes in ideal KTO are **consistently observed and attributed to PNRs** [X. Wen et al. *Appl. Phys. Lett.* 122, 232905 (2023); S. Kojima et al. *Jpn. J. Appl. Phys.* 57, 11UB05 (2018); S. Glinsek et al. *J. Appl. Phys.* 111, 104101 (2012); H. Uwe et al. *Phys. Rev. B* 33, 6436-6440 (1986); O. Aktas et al. *Phys. Rev. B* 90, 165309 (2014)].

(2.2) Raman spectroscopic measurements

(2.2.1) Experimental results

We conducted temperature-dependent Raman spectroscopy on LAO/KTO(111) heterostructures (both before and after electrostatic poling), using $\text{KTa}_{0.65}\text{Nb}_{0.35}\text{O}_3$ ($\text{KTN}_{0.35}$)(111) and KTO(111) single-crystal substrates as controls (Fig. R1). Electrostatic poling was performed at -180 V and 4.5 K in a vacuum cryostat. Following poling, the sample was rapidly warmed to room temperature for Raman measurements to minimize relaxation effects (Supplementary Fig. 3 confirms that brief ambient exposure does not fully erase the poled state).

As shown in Fig. R1a, the commercially available ferroelectric $\text{KTN}_{0.35}$ (111) single crystals exhibit prominent TO_2 and TO_4 modes, indicative of symmetry breaking. These modes persist up to ~ 175 K, confirming the ferroelectric character. In the KTO(111) substrates, weaker TO_2 ($\sim 198 \text{ cm}^{-1}$) and TO_4 ($\sim 544 \text{ cm}^{-1}$) peaks are observed until ~ 40 K (Fig. R1b), consistent with previous studies attributing them to PNRs. Crucially, the LAO/KTO(111) heterostructures exhibit similar features (Fig. R1c), with the TO_4 peak persisting to ~ 60 K post-poling (Fig. R1d).

To enhance the visibility of these features, we performed background subtraction (using high-temperature references) and data normalization. The processed TO_4 peak intensities versus temperature are shown in Fig. R1e. At 60 K, the poled sample shows a distinguishable TO_4 peak (Fig. R1f), absent in the unpoled case; at 1.6 K, the TO_4 peak is stronger in the poled sample than the unpoled one (Fig. R1g). The enhanced intensity and elevated T^* demonstrate improved PNR alignment after poling.

(2.2.2) Conclusions

(i) PNR-induced weak ferroelectricity exists in both KTO substrates and LAO/KTO heterostructures. This polarization originates primarily from the bulk KTO substrate (not exclusively interfacial).

(ii) Switchable R_s serves as a polarization indicator. The temperature range of hysteresis in R_s - V_G occurs matches the Raman-derived T^* where the transverse polar optic phonon modes (TO_2 and TO_4) appear, confirming R_s as a reliable ferroelectric probe—consistent with reports on Ca-doped SrTiO_3 interfaces [*J. Brehin et al. Phys. Rev. Mater.* **4**, 041002(R) (2020); *P. Noël et al. Nature* **580**, 483–486 (2020)].

These results, combined with transport data, provide conclusive evidence for ferroelectricity in LAO/KTO.

(3) Connecting PNR-induced substrate ferroelectricity to the interfacial ferroelectricity coexisting with superconductivity

In the following discussion, we elucidate how *switchable* substrate ferroelectricity enables the *switchable* interfacial ferroelectricity in LAO/KTO heterostructures:

(i) The conductive interfacial KTO layer remains crystallographically continuous with the bulk substrate. While E -fields cannot directly polarize this conductive layer, gating through insulating bulk KTO modulates bulk polarization, which

coherently alters interfacial polarization (*enabling switchable ferroelectric control of superconductivity*).

- (ii) The ferroelectric tuning of interfacial conduction *cannot* be explained by conventional capacitance effects (which would require only bulk KTO ferroelectricity). Beyond arguments in the main text, we emphasize that PNR-induced ferroelectric polarization—as established in previous studies—is exceptionally weak ($\sim 0.04 \mu\text{C}/\text{cm}^2$), ruling out purely capacitive coupling.
- (iii) **Importantly**, we clarify a subtle but crucial point: While the interfacial KTO layer’s polarization may differ quantitatively from bulk KTO’s, their inherent lattice bonding ensures that bulk polarization switching induces interfacial switching.

Revision Summary:

(1) We have added Raman scattering measurements as Supplementary Note 1 and Supplementary Fig. 4 in the revised manuscript. The corresponding text in the main text now reads (Page 5):

“Further evidence for low-temperature ferroelectricity in LAO/KTO comes from our Raman scattering measurements (Supplementary Fig. 4), where the TO_2 and TO_4 optical modes emerge below ~ 40 K (~ 60 K in poled samples) – closely coinciding with the onset of hysteresis in the V_G - R_s loops (Figs. 2g-h).”

(2) On page 4, we have rewritten the last sentence in the “**Ferroelectric hysteresis under gating cycles**” section as:

“This temperature-dependent behavior closely matches previous observations^{39–41} of weak spontaneous polarization ($0.04 \mu\text{C}/\text{cm}^2$) in KTO, where defect-induced polar nanoregions develop macroscopic coherence within this specific temperature range.”

(3) On Page 5, we have added the following discussion paragraphs:

“Notably, even nominally pure KTO samples typically contain substantial defect concentrations ($\sim 10^{17} \text{cm}^{-3}$). These defects promote the formation of polar nanoregions, which collectively drive KTO into a weak ferroelectric state at low temperatures^{40,46}.”
and

“The similarities between the observed hysteresis and Raman features in LAO/KTO and those in the intentionally ferroelectric $\text{Al}/\text{Sr}_{0.99}\text{Ca}_{0.01}\text{TiO}_3$ interface²⁹ provide additional support for ferroelectricity in LAO/KTO.”

(4) On Page 6, we have refined the concluding statement in the “**Coexistence of ferroelectricity and superconductivity in the conducting layer**” section to read:

“Although ferroelectricity and conductivity are traditionally considered mutually exclusive, polarization in the conducting KTO layer *could emerge via coupling with the ferroelectric KTO bulk, mediated by their shared lattice bonding, while potentially developing distinct interfacial behavior.*”

Another major issue I found that the authors have completely ignored the issue of charge trapping/detrapping under electrical gating. This has been demonstrated extensively in context of SrTiO₃ based heterostructures [Sci. Rep. 4, 6788 (2014); Phys. Rev. Lett. 124, 017702 (2020); Phys. Rev. Applied 15, 054008 (2021)]. Such charge trapping phenomena also gives rise to hysteresis in resistance vs. gate voltage measurement. Similar observations have been reported in Fig. 2 of this manuscript but attributed to the ferroelectricity. The authors need to demonstrate that their observations are not linked with charge trapping phenomena.

Reply: We thank the reviewer for raising this important point regarding charge trapping effects, as documented in SrTiO₃-based systems [Sci. Rep. 4, 6788 (2014); Phys. Rev. Lett. 124, 017702 (2020); Phys. Rev. Appl. 15, 054008 (2021)]. Below, we present four key lines of evidence and reasoning that indicate charge trapping is unlikely to be the dominant mechanism in our LAO/KTO results.

Line 1: Distinct phenomenological differences

To better address the reviewer's concern, we performed additional experiments to systematically trace the evolution of R_s with V_G under continuous V_G sweeps. The results are summarized in Fig. R2.

Fig. R2 (also presented as **Supplementary Fig. 5** in the revised manuscript) | **Ferroelectric-like hysteresis independent of first sweep polarity in back-gated LAO/KTO.** a-c, V_G - R_s characteristics of LAO/KTO heterostructures exhibit stable bistable states (a) and a systematic increase in hysteresis width with increasing maximum gate voltage ($V_G^{\max} = 60, 120, 180$ V), irrespective of the polarity of the initial voltage sweep (b, c). The hysteresis loops ultimately stabilize into a reproducible double-state configuration at zero bias (0^{\pm}), consistent with robust ferroelectric polarization switching behavior. FS: forward sweep; BS: backward sweep.

For comparison, we reproduced in Fig. R3 the typical R_s - V_G curves from previous studies on SrTiO₃-based systems [adapted from Phys. Rev. Lett. 124, 017702 (2020)] and Sci. Rep. 4, 6788 (2014)].

Fig. R3 Representative R_s - V_G behavior from SrTiO₃-based interfaces under (a) first forward and (b) first backward sweeps. Adapted from Refs. [J. Biscaras *et al. Sci. Rep.* 4, 6788 (2014); C. Yin *et al. Phys. Rev. Lett.* 124, 017702 (2020)].

We can clearly see that the R_s - V_G curves for our LAO/KTO heterostructure are very different from those of SrTiO₃-based systems, exhibiting three features **inconsistent with charge trapping but consistent ferroelectric hysteresis**.

(i) Reproducible and switchable hysteresis

- *Observation:* After the initial forward sweep, we observe a pronounced, *repeatable* hysteresis in the R_s - V_G curve.
- *Contrast with SrTiO₃:* In SrTiO₃-based heterostructures, charge trapping causes an irreversible resistance change during the *first* sweep. Subsequent sweeps (for $V_G < V_G^{\max}$) exhibiting *overlapping trajectories*.

(ii) Independent of initial V_G direction

- *Observation:* Regardless of whether the initial V_G is positive or negative, the system stabilizes into a *robust hysteretic loop*.
- *Contrast with SrTiO₃:* SrTiO₃-based heterostructures exhibit *direction-dependent* reversibility (e.g., irreversible forward sweeps vs. reversible negative sweeps).

(iii) Non-volatile binary states at zero voltage

- *Observation:* The hysteresis stabilizes into two well-defined R_s states at $V_G = 0$ (e. g., 0^\pm states), a hallmark of ferroelectric polarization switching.
- *Contrast with SrTiO₃:* Charge trapping in SrTiO₃ does not produce such deterministic, non-volatile states at zero bias.

These behaviors are signatures of ferroelectricity and cannot be explained by charge trapping mechanisms reported in SrTiO₃-based systems.

Line 2: Striking parallels with *ferroelectric* (Sr,Ca)TiO₃-based heterostructure

While SrTiO₃-based systems typically exhibit negligible hysteresis, *ferroelectric* (Sr,Ca)TiO₃ interfaces (with isovalent Ca²⁺ substitution) demonstrate clear switchable two-dimensional electron gas and hysteresis in R_s - V_G curves [J. Brehin *et al. Phys. Rev.*

Mater. 4, 041002(R) (2020); P. Noël et al. *Nature* 580, 483–486 (2020)]. Remarkably, our LAO/KTO interface reproduces this hysteretic behavior, including the same characteristic loop shape and non-volatile states. The similarity to this known ferroelectric interface provides compelling evidence for ferroelectric modulation in LAO/KTO.

Line 3: Carrier concentration stability

In charge trapping scenarios, electron trapping leads to substantial carrier density reduction after bias removal [e.g., up to 43% (*J. Biscaras et al. Sci. Rep.* 4, 6788 (2014)) or even inducing insulating behavior (*C. Yin et al. Phys. Rev. Lett.* 124, 017702 (2020))]. In striking contrast, our LAO/KTO heterostructures retain nearly unchanged carrier concentration in the "0+" state, with some samples even showing slight increases (Figs. 3c-d; Supplementary Fig. 1b) - demonstrating mobility-dominated behavior rather than carrier density changes, which further contracts charge trapping mechanisms.

We note that KTO's dielectric constant ($\epsilon \approx 5000$ at low T) is only $\sim 25\%$ of SrTiO₃'s, fundamentally limiting its carrier density modulation capability. Even under upper-bound estimates, a 180 V V_G variation would induce $\Delta n \leq 1 \times 10^{13} \text{ cm}^{-2}$ (with actual values further suppressed by field-induced ϵ suppression).

Together, these experimental observations and material constraints provide further compelling evidence against charge trapping as the dominant mechanism.

Line 4: Temperature-dependent hysteresis correlates with Raman spectra

The R_s - V_G hysteresis emerges only below ~ 50 K (Fig. 2), mirroring the onset of transverse polar optic modes observed by Raman spectroscopy (Fig. R1). This synchronized temperature dependence between transport and spectroscopic data strongly implicates ferroelectricity as the dominant mechanism.

Revision summary:

We have incorporated Fig. R2 as Supplementary Fig. 5 in the revised manuscript. Correspondingly, we have added the following discussion paragraph on Page 6:

“While V_G -induced irreversibility in R_s has been previously observed in SrTiO₃-based heterostructures and attributed to charge trapping/detrapping effects⁴⁷⁻⁴⁹, the behavior we observe in LAO/KTO is fundamentally distinct. Our system exhibits reproducible and switchable hysteresis that is independent of the initial V_G sweep direction – a hallmark of ferroelectricity. Moreover, the limited magnitude of carrier density modulation in LAO/KTO is incompatible with charge trapping, which would require significantly larger n_s variations^{47,48}. Taken together, the observed hysteresis and switching behaviors in LAO/KTO cannot be explained by charge trapping or ionic migration, providing compelling evidence for ferroelectricity as the governing mechanism.”

Because of these two significant issues in the manuscript, I do not recommend the

publication of this paper in Nature Communications.

Reply: We sincerely appreciate the reviewer's constructive critique which has substantially improved our manuscript. To address the two key concerns raised:

1. We have performed new experimental characterizations (Raman spectroscopy and systematic transport measurements) that provide compelling support for our conclusions.
2. We have thoroughly revised our discussion to deliver more rigorous analysis and strengthened interpretations.

While we understand the reviewer's initial reservations, we believe the enhanced experimental evidence and improved discussion now comply with the journal's rigorous criteria. We are truly grateful for the reviewer's insightful comments and time spent evaluating our work and hope the revised manuscript merits reconsideration for publication.

Responses to the comments from Reviewer #1

I have read the authors' response and the revised manuscript. I am impressed by the scientific care and scrutiny the authors show in addressing all concerns and questions raised in the first round of review. In general, I am satisfied with the responses and revisions. The manuscript has been extensively improved, particularly with the scientific significance clearly stressed and validity of main conclusions strengthened by the newly-supplied Raman data. I also appreciate that the authors mentioned Feigel'man's fractal superconductivity as a potential origin of T_c enhancement. Taking all these together, my recommendation would be to publish this work in *Nature Communications*.

Reply: We sincerely thank the reviewer for the positive assessment of our work and greatly appreciate the time and effort devoted to improving the quality of this manuscript.

Meanwhile, I have one remaining suggestion. As elucidated by the authors in the reply file, the transport coefficients derived from the Hall measurement, including n_s and μ , are weighted spatial average of all contributing layers since these parameters are inherently dependent on the depth along z direction. This point is not mentioned in the manuscript, while I think it is helpful to the reader to see it addressed. Probably a brief note at the end of the paragraph introducing these parameters (the first paragraph on page 5) can resolve this matter. Also, n_s and μ are only defined in the Methods but not in the main text.

Reply: We thank the reviewer for these constructive suggestions. To address these concerns, we have made the following revisions:

(1) We have added the following clarification at the end of the first paragraph on page 5: "It should be noted that the transport coefficients, n_s and μ , derived from Hall measurements on Hall bar devices, represent spatially weighted averages of all contributing layers, as these parameters are inherently depth-dependent along the z -direction."

(2) We have explicitly defined n_s and μ in the first paragraph of page 5: "...the increase in T_c was accompanied by a clear decrease in mobility μ (with a slight increase in carrier density n_s ; however, the change in n_s was much smaller than that of μ) ..."

Responses to the comments from Reviewer #2

I have gone through the revised manuscript along with the response letter and supplemental materials. I am still not convinced by the claim about the ferroelectricity. The authors have performed Raman spectroscopy, and the polar mode TO_2 and TO_4 are present in every sample they examined, including a bare KTO (111) substrate. While the mode confirms the presence of polar distortion, it does not confirm ferroelectricity. The polar distortion may not be uniform and can be confined within a few unit cells. It is well known that depending on the concentration of polar nanoregions and temperature, one can get several phases, including paraelectric, ferroelectric, and dipolar glass phase [see *J. Phys. Condens. Matter* 15, R367–R411(2003)]. In fact, in one recent article [*Nature Communications* 15, 3830 (2024)], it has been demonstrated the electron dynamics of metallic $KTaO_{3-x}$ sample behaves like a glass, which is related to the dipolar glassy phase of polar nanoregions. Overall, the experimental results reported in this paper do not confirm ferroelectricity, although that may be a possibility. So, I do not recommend the publication of this manuscript in *Nature Communications* in its present form.

Reply: We thank the reviewer for their detailed comments and for acknowledging the evidence of polar distortions. We understand the reservation regarding the confirmation of ferroelectricity and offer the following clarification to align our interpretation with the reviewer's perspective.

Firstly, we fully agree that the Raman modes indicate polar distortions which could correspond to various phases, including a dipolar glass as the reviewer rightly points out. Our claim is not that the material is intrinsically ferroelectric in its pristine state.

Secondly, our key argument is that the application of an electric field aligns these polar nanoregions (PNRs), inducing a switchable ferroelectric state. The hysteresis loops we observe are direct evidence of this field-induced ferroelectricity. We believe this effect occurs primarily within the insulating bulk KTO substrate, as this is the region across which the gate voltage is effectively applied. The ferroelectric properties observed at the conducting interface are thus a consequence of this effect propagating from the adjacent bulk region.

Therefore, our results demonstrate how the inherently disordered PNRs in KTO can be electrically manipulated to exhibit functional and switchable ferroelectric properties.

To address the reviewer's concerns directly, we have added a clarifying statement at the end of the first paragraph on page 6: “We attribute the induced ferroelectricity to a field-induced ordering of the polar nanoregions that are present in KTO, which are inherently disordered and lacking long-range correlation in the pristine state.”